# Development of an HPLC-MS/MS Method for the Determination of Alkaloids in Lupins

**DOI:** 10.3390/molecules28041531

**Published:** 2023-02-05

**Authors:** Fabiola Eugelio, Sara Palmieri, Federico Fanti, Luana Messuri, Alessia Pepe, Dario Compagnone, Manuel Sergi

**Affiliations:** 1Department of Bioscience and Technology for Food, Agriculture and Environment, University of Teramo, Via Renato Balzarini 1, 64100 Teramo, Italy; 2Department of Chemistry, Sapienza University of Rome, Piazzale Aldo Moro 5, 00185 Rome, Italy

**Keywords:** alkaloids, HPLC-MS/MS, solid-phase extraction, lupins, *Lupinus albus* L.

## Abstract

Lupin alkaloids (LAs) represent a class of toxic secondary metabolites in plants, in particular in *Lupinus* spp.; they are produced as a defense mechanism due to their strong bitter taste and are very dangerous for human and animals. In this work, a sensitive and reliable high performance liquid chromatography—tandem mass spectrometry (HPLC-MS/MS) analytical method for the identification and quantification of thirteen lupin alkaloids was developed and validated according to FDA guidelines. Efficient extraction and clean-up steps, carried out by solid-phase extraction, were finely tuned on the basis of the characteristics of the analytes and lupin samples, providing good selectivity with minimized matrix interference. The effectiveness of the method was proven by the satisfactory recovery values obtained for most of the analytes and a matrix effect ≤23% for all tested levels. In addition, a sensitive and reliable determination of the target compounds was obtained; LOQs were between 1 and 25 µg Kg^−1^, i.e., below the requested maximum levels (<200 mg Kg^−1^). The method was applied to evaluate the LAs profile in different batches of raw *L. albus* L. samples, varying in size and across farming treatments.

## 1. Introduction

Alkaloids are secondary metabolites present in different plant species, representing one of the most heterogeneous classes of compounds. They are produced as a defence mechanism against insect pests, pathogens, herbivorous animals, and competing plants due to their typical bitter taste and toxic effects [1].

In the legume family (Leguminosae/Fabaceae), more than 170 chemical structures belonging to this class of compounds were found, occurring particularly in the lupin species (*Lupinus* spp.) [2]; for this reason, they are commonly referred as “lupin alkaloids” (LAs) [3]. They are also notable in the genera *Baptisia, Thermopsis, Genista, Cytisus, Laburnum,* and *Sophora* [2].

LAs are mainly represented by quinolizidine alkaloids (QAs), a toxic alkaloid class produced from L-lysine that is converted to cadaverine, the main intermediate of this class of compounds [4]. They are biosynthesized in the green tissues of the plant, transported via the phloem and stored in all organs, seeds included [5]. QAs may be present in bicyclic, tricyclic, and tetracyclic forms [6] with a core structure (C_5_NC_4_ skeleton) characterized by two fused 6-membered rings with a nitrogen atom at the bridgehead, as shown in Figure 1a [4]. More than 70 QAs have been found in lupin species, specifically the most relevant being lupanine, sparteine, and lupinine; levels and combinations of QAs have been reported to vary according to different factors, such as species or variety, botanical and geographic origin, but also to soil composition and climate [7]. Alkaloids with an indole as the core group (Figure 1b), such as gramine, can also be found in lupin seeds.

It is possible to distinguish two main types of lupins, sweet and bitter varieties, which differ in the alkaloid content [4]. In the first type, developed by plant breeders, the alkaloid amount ranges between 100 and 800 mg Kg^−1^ of dry weight; consumption by humans and animals is allowed [8]. Bitter lupins are characterized by a higher concentration of alkaloids, usually between 5000 and 40,000 mg Kg^−1^ of dry weight [9].

LAs can cause symptoms of poisoning in humans, affecting the nervous, circulatory, and digestive system [6]; for this reason, bitter lupin seeds are not suitable for human or animal consumption without a proper pre-treatment [7]. In fact, a debittering process involving boiling and then soaking or washing under running water for several days is needed [10].

Recently, lupin seeds have received considerable international interest as a human health food ingredient due to their high protein and dietary fiber contents [11], but also for their low glycemic impact and gluten-free characteristics [12]. According to this, regulations in Australia, New Zealand, France, and Great Britain require compliance with a maximum level of 200 mg Kg^−1^ of alkaloids in lupin flours and seeds [13].

The species that are mainly considered in Europe are *Lupinus angustifolius* L. (narrow-leaf or blue lupin), *Lupinus albus* L. (white lupin), *Lupinus luteus* L. (yellow lupin), and *Lupinus mutabilis* L. (Andean lupin) [12]. Among them, *Lupinus albus* L. (*L. albus* L.), considered in this work, is an old leguminous plant that can grow in different soils and climates; moreover, it is consumed particularly around the Mediterranean area [14]. It has great potential as a crop plant due to a high seed yield as well as its protein and oil contents in seeds [15]. In this species, lupanine is the most abundant alkaloid together with other LAs such as albine, 13-hydroxylupanine, isolupanine, angustifoline, 13-angeloyloxylupanine [2], N-methylalbine, and some esters of 13-hydroxylupanine [16].

Several analytical methods have been reported for the identification of the alkaloid profile in seeds and other plant parts of different lupin species by using capillary gas- chromatography coupled with mass spectrometry (GC-MS) [17,18,19]. More recently, other methods were developed by reversed-phase liquid chromatography interfaced with electrospray ionization coupled with high-resolution MS (RPLC-ESI-HRMS), mainly using a quadrupole-time of flight (Q-TOF) or Orbitrap MS instruments. The latter provides high selectivity and sensitivity for the untargeted screening of phytotoxins, including some QAs such us lupinine, sparteine, and cytisine and indole alkaloids such as gramine, in environmental samples, lupins, feed, and plant extracts [20,21,22,23].

A study based on the characterization of the metabolites in the active fraction of *L. albus* L. seeds was developed using ultra-high-performance LC–tandem mass spectrometry (UHPLC-MS/MS) analysis, performed by a hybrid quadrupole-orbitrap mass spectrometer [24]. LAs were also investigated in tissue sections of seed-containing pods of *Lupinus angustifolius* L. (*L. angustifolius* L.), using matrix-assisted laser desorption ionization imaging (MALDI-MSI) [25].

Although different approaches are presented in the literature, according to the German Federal Institute for Risk Assessment (BfR) Opinion No 003/2017, only of a few methods for the quantification of the alkaloids in lupins and derived foods are described and validated in the literature because of the limited availability of standard substances [7]. Most of these methods are mainly based on GC-MS [5,8,13,16,26] and GC with flame ionization detection (GC-FID) [27,28]. A study that aimed to determine the LAs in different *Lupinus* species by non-aqueous capillary electrophoresis (NACE) by using ultra-violet (UV) and MS detection was also reported [29].

In recent years, there has been an increase in the development of targeted methods by LC-MS/MS to detect LAs in various matrices, such as environmental [30] and biological samples; for example, sparteine and lupanine were detected and quantified among other alkaloids in the blood for the examination of possible poisoning of humans by plants [31], while Green et al. used an LC-MS method to quantify the major LAs of *Lupinus leucophyllus* L. (anagyrine, lupanine, and 5,6-dehydrolupanine) in bovine serum [32]. Anyway, these are not fully suitable LC-MS/MS methods for the quantification of LAs in lupin [4,7], particularly in *L. albus* L.; Engel et al. developed a method for the quantification of LAs in the milk of cows fed seeds of *L. angustifolius* L. and in the feed itself [33]. To the best of our knowledge, only a recent study developed and validated an LC-MS/MS method for the rapid and simultaneous quantification of lupinine, 13-hydroxylupanine, lupanine, angustifoline, and sparteine in the raw seeds of *L. angustifolius* L. and lupin-processed products [34].

Considering the increasing interest of consumers, both in Europe and in other countries, for lupins as food ingredients and snacks, thanks to their nutritive characteristics, the aim of this work was the development of a sensitive targeted method involving the use of HPLC-MS/MS, operating in multiple reaction monitoring (MRM) mode, for the simultaneous quantification of thirteen different LAs. The development of a fast and efficient extraction procedure followed by a reliable clean-up step was performed by Solid Phase Extraction (SPE) to remove the interfering compounds and to obtain satisfactory recoveries.

The presented analytical method was validated following the Food and Drug Administration (FDA) guidelines [35] and was applied to different batches of raw *L. albus* L. samples, varying in size and treatments, in order to quantify and assess their LA content.

## 2. Results and Discussion

### 2.1. Extraction Procedure Development

In the literature, most of the studies focused on a specific alkaloid extraction procedure, based on the dissolution in acidic solution and subsequent alkalinization, followed by a further purification by means of liquid–liquid extraction or SPE [36]. In this work, the extraction procedure was studied in order to obtain rapid, efficient, and selective recovery of all analytes from the lupin matrix, exploiting the use of a small amount of sample and solvents. Different ratios of MeOH:H_2_O and ACN:H_2_O (*v*:*v*) in 1 mL were tested on 200 mg of sample, as reported in Figure 2a. Different aliquots of the same homogenized sample were extracted with different ratios of solvents (n = 5), and the chromatographic area was compared. The maximum extraction yield was obtained using a solution of MeOH:H_2_O 60:40 (*v*:*v*).

### 2.2. Clean-Up Procedure Development

The clean-up step was tuned by developing an efficient SPE procedure to provide suitable recoveries and the minimization of the matrix effect for the analytes. For this purpose, two different SPE cartridges were initially evaluated, considering different types of stationary phases, such as a silica-based Strata C18-E (55 μm particle size) and a polymeric Strata XL (100 μm particle size); the different stationary phases were selected for their different mechanisms of interaction with the target analytes to achieve a satisfactory clean-up procedure.

The C18 cartridges are silica-based, characterized by hydrophobic selectivity, while the Strata XL is a polymeric-based material with a styrene–divinylbenzene sorbent bed, characterized by three retention mechanisms (π–π bond, hydrogen bond, and hydrophobic interaction). The strata-C18 cartridges provided better retention for all analytes, even with high amounts of organic solvent, but did not guarantee good recoveries in the elution step in term of the compound amount and reproducibility. For this reason, Strata-XL cartridges were selected. To evaluate the best conditions for the clean-up step, different tests were performed.

The first step was characterized by the selection of the loading phase to obtain a satisfactory retention of the target analytes. Different ratios of H_2_O:MeOH (*v*:*v*) were tested (90:10, 80:20, 70:30, 60:40, 50:50 *v*:*v*), and the analytes were evaluated by comparing the area of the single unretained analyte to the area of a reference mixture with known concentration. The best conditions were obtained using H_2_O:MeOH 90:10 (*v*:*v*) that allowed the retention of the majority of the analytes in the cartridge, as shown in Figure 2b; the selected condition was then applied to the evaluation of washing and elution steps. Moreover, for the washing step, 100% H_2_O and 90:10 and 80:20 ratios (*v*:*v*) of H_2_O:MeOH were tested; H_2_O:MeOH 90:10 (*v*:*v*) was selected as the washing solution due to the reduced amount of lost compounds and, at the same time, low organic solvent was used. The latter was necessary to remove interferents affecting the ionization step in LC-MS/MS; this is reported in Figure 2c.

After the set-up of the previously mentioned step, the elution conditions were tuned, testing MeOH and different ratios of MeOH:H_2_O (90:10, 80:20, 70:30, 60:40, 50:50 *v*:*v*). As reported in Figure 2d, 100% MeOH allowed a reliable recovery for all of the target compounds from the cartridge (>50%). Despite that the presence of H_2_O in different tested solvents improved the recovery of some analytes (i.e., lupanine, thermopsine, and angustifoline), it negatively affected the elution of others (i.e., N-formylcytisine, N-methylcytisine, cytisine, and lupinine); indeed MeOH was a good compromise to obtain a good recovery for all target analytes.

### 2.3. HPLC−MS/MS Method Development

Tandem mass spectrometry parameters were carefully tuned by direct infusion of each analytical standard at a concentration of 10 ng mL^−1^ and flow rate of 10 µL min^−1^. The most intense fragments were selected for each compound according to the literature [30,31,34,37] and tested during chromatographic runs; then, the two most intense MRM transitions based on the S/N ratio were selected as the quantifier and qualifier.

The chromatographic run was set up to obtain a good separation of compounds and an acceptable peak shape, minimizing fronting and tailing effects, which can interfere with the proper quantification of analytes, and to obtain the highest possible sensitivity.

To achieve this goal, different stationary phases were tested: a mixed ligand C18-pentafluorophenyl (PFP) Ace Excel 2 (100 mm × 2.1 mm) packed with 2 μm particles, an Atlantis dC18 (50 × 2.1 mm) packed with 3 μm particles, and a Phenomenex Kinetex PFP column (75 × 3 mm) packed with 2.6 μm particles. The aim was to evaluate the effect of different types of interaction; the C18 phase provides hydrophobic selectivity; on the other hand, the PFP phase promotes several interactions—particularly relevant in this case is hydrogen bonding, which allows the interaction with the nitrogen moiety in the heterocyclic ring, improving the retention.

During the development of the LC method, the initial test was performed on a C18 stationary phase that gave a good separation of the analytes, except for more polar compounds that were poorly retained. On the other hand, the PFP stationary phase allowed a better retention for all compounds but provided a lower separation efficiency. The combined C18-PFP stationary phase appeared the most suitable for the analytes of interest, providing good performances for the retention, peak shape, and selectivity for all compounds. This combined stationary phase maintained the hydrophobic interaction and low bleed characteristics of the C18 phases, with an improved retention of polar compounds due to the PFP phase.

Several combinations of mobile phases containing different amounts of formic acid (ionization enhancer) were initially tested; good performances were obtained using water (H_2_O) with 0.1% formic acid as phase A and acetonitrile (ACN) as phase B, as reported in previous work [38]. However, peak tailing was still present, particularly for sparteine, in addition to the peak splitting of cytisine and N-methylcytisine.

With the aim of untangling these issues, formic acid was replaced with heptafluorobutyric acid (HFBA), which was added in both the A and B phase at 0.1%; this improved both the peak shape and ionization yield. Phase B was then replaced with ACN:MeOH 50:50 (*v*:*v*) 0.1% HFBA for further enhancement of ionization. The resulting chromatographic profile was satisfactory, as reported in Figure 3. The extracted ionic currents (XIC) for each analyte are reported in Figure 4.

### 2.4. Validation Results

Validation parameters were then evaluated according to FDA guidelines [35]. All QCs samples were prepared in lupins homogenate as reported in Section 3.5.1; briefly, different QCs were prepared at different concentrations (25, 50, 500 µg Kg^−1^), and replicates (n = 5) were tested. Moreover, a 10-point calibration curve was prepared in MeOH (0.1, 0.25, 0.5, 1, 2.5, 5, 10, 25, 50, 100 ng mL^−1^) for the linearity test. Good linearity was achieved for each analyte, with coefficients of determination (r2) > 0.99 as reported in Table 1. Moreover, the limits of determination and quantification (LODs and LOQs) were lower than those obtained for the majority of methods reported in the literature (Table 1), allowing the detection and quantification of alkaloids in lupin matrices even in trace amounts. The LOD and LOQ values were calculated as described in Section 3.5.2, proving the capability of the method to detect analytes under the requested maximum level (<200 mg Kg^−1^).

In addition, satisfactory recoveries for most of the analytes were obtained as shown in Appendix A. A matrix effect ≤23% was obtained for all tested amounts (Figure 5a), confirming the effectiveness of the developed clean-up procedure for the removal of the interfering compounds. The accuracy and precision of the method were also tested, which, as shown in Figure 5b,c, are in line with the guidance provided by the FDA [39]. Moreover, no carry-over effects were observed during the tests.

### 2.5. Analysis of LAs in Lupinus albus L. Samples

The developed method was applied to four different batches of raw seeds belonging to *L. albus* L. species from the Abruzzo region supplied by the Madama Oliva srl Company, with the aim to evaluate the LA content in different types of lupin samples. Indeed, three different sample batches were subjected to conventional farming and were characterized by a different caliber (11/13, 13/15 and 15/17 mm); on the other hand, a batch of the same supplier was from organic farming (caliber: 15/17 mm). Each batch was analyzed in triplicate with the developed analytical procedure; the results are reported in Appendix A.

First, it was possible to observe a high amount of lupanine in all samples, the most predominant alkaloid in *L. albus* L., as already known in literature; the other determined LAs were multiflorine, albine, hydroxylupanine, angustifoline, and sparteine as reported in other work [17]. In a preliminary evaluation of the data, it was clear that the total alkaloid content decreased as the seed’s caliber increased.

In order to test the significance of differences between organic and conventional farming samples, but also between the smaller-sized sample (11/13 mm) and the other samples, one-way ANOVA with a parametric test and multiple comparison option was performed at a single LA level. In Figure 6, the comparison between different lupin calibers is shown; in this case, a significant decrease in the alkaloid content, in conjunction with the increase in the seed size in conventional farming samples can be observed; in particular, lupanine was up to 3-fold more concentrated in smaller lupins, with *p* ≤ 0.001, and hydroxylupanine was up to 4-fold more concentrated, with *p* ≤ 0.001. As regards other LAs, such as sparteine and angustifoline, they had a concentration increase in smaller lupins too, with a *p*-value ≤ 0.005 for both. On the other hand, multiflorine and albine did not show a significant decrease in their concentration correlated with different seed sizes. Therefore, the 11/13 mm-sized lupins appeared to have an LA content higher than that of the 13/15 and 15/17 mm-sized lupins, corresponding to commercial samples. This information may be relevant to obtain an LA content lower than 200 mg Kg^−1^ in the commercial samples. Moreover, this is in accordance with the already known literature about the correlation of LAs and growth time; this is caused by the expression of lysine decarboxylase that leads to the formation of the main precursor of LAs [25].

Moreover, the same statistical analysis was performed to evaluate differences in the LA content between conventional and organic farming seeds with the same size. In this case, an increase in LAs in the conventional sample of the same size, with a significant increase in lupanine (up to 4-fold in the organic farming samples with a *p*-value ≤ 0.001), was observed (Figure 7). In the organic samples, hydroxylupanine was significantly higher than that in other samples, (3-fold higher with a *p*-value ≤ 0.005). The increase in LAs, most likely, was dependent on the use of nitrogen fertilizers commonly applied in organic farming [40]. In addition, it is known that the lack of phytochemicals may improve the synthesis of alkaloids due to bactericide-like effects of LAs [41].

According to these preliminary data, the different sizes and farming conditions appeared to have a significant influence on the LA content in *L. albus* L. This information can be useful in farming, leading to the selection of the conditions for the industrial debittering process, a necessary step for lupin edibility.

## 3. Materials and Methods

### 3.1. Chemicals and Samples

The alkaloid standards used in this research were N-formylcytisine and N-methylcytisine from LGC Standards Srl (Milano, Italy); (+)-lupanine (perchlorate), 13α-hydroxylupanine, and thermopsine, purchased from Vinci-Biochem Srl (Firenze, Italy); gramine, cytisine, (+)-sparteine, and (−)-lupinine, purchased from Merck Life Science Srl. (Milano, Italy), (−)-angustifoline, multiflorine, and albine (hydrochloride), purchased from Phytoplan (Heidelberg, Germany), and rhombifoline, purchased from Clinisciences (Guidonia Montecelio, Italy). The chemical structures, CAS number, and other specific information of these standards are reported in Appendix A.

The working standard mixtures were prepared by appropriate dilutions in MeOH (10, 1, 0.01 µg mL^−1^) with storage at −20 °C. HFBA and formic acid were purchased from Merck Life Science Srl (Milan, Italy). H_2_O, MeOH, and ACN, all UPLC-MS grade, were purchased from VWR (Radnor, PA, USA).

### 3.2. Sample Extraction

Different categories of the treated lupin samples were, initially, homogenized by the use of a blender. Then, 200 mg of homogenized lupin samples was weighed and extracted with 1 mL of MeOH:H_2_O 60:40 (*v*:*v*) by a Precellys Evolution homogenizer (Bertin Technologies SAS, Montigny-le-Bretonneux, France) with 3 cycles of 10 sat 7000 rpm with a 45 s stop between each cycle followed by centrifugation for 10 min at 4 °C at 11424 RCF. Then, 50 µL of the supernatant was collected and diluted in 1 mL with a final ratio of H_2_O:MeOH 90:10 (*v*:*v*). The resulting solution was loaded onto a polymeric SPE cartridge for the clean-up step.

### 3.3. Clean-Up

The clean-up step was carried out by SPE, using Strata-XL 100 µm Polymeric Reversed Phase cartridges (330 mg mL^−1^) from Phenomenex (Torrance, CA, USA). They were first conditioned with 1 mL of MeOH and subsequently equilibrated with 1 mL of H_2_O:MeOH 90:10 (*v*:*v*). Each extract was diluted in 1 mL of the equilibrating solution and loaded onto the cartridge. For the washing step, 1 mL of H_2_O:MeOH 90:10 (*v*:*v*) solution was used to remove the interfering compounds. Finally, the analytes were eluted with 1 mL of MeOH and collected for HPLC-MS/MS analysis.

### 3.4. HPLC–MS/MS Analysis

The analysis of alkaloids was performed by a Nexera XR LC system (Shimadzu, Tokyo, Japan) coupled to a 4500 QTrap mass spectrometer (Sciex, Toronto, ON, Canada) equipped with a heated ESI source. The analytes were separated using an Excel 2 C18-PFP (10 cm × 2.1 mm ID) column from ACE (Aberdeen, UK), packed with 2 µm particles and equipped with a security guard column.

The mobile phases consisted of H_2_O with 0.1% of HFBA (A) and ACN:MeOH 50:50 (*v*:*v*) with 0.1% of HFBA (B). The elution of the analytes was carried out with the following gradient: start with 10% of phase B for 0.1 min; linear increase at 50% of B in 3 min; isocratic step at 50% of B for 4 min; linear increase at 90% of B in 3 min; isocratic step at 90% of B for 0.5 min; back to the initial conditions (10% of B) in 0.5 min. The injection volume and the flow rate were set at 6 µL and 0.300 mL min^−1^, respectively. The separation of the analytes took place in 10 min, and the total duration of the analysis was 12 min, including the 2 min rebalancing of the initial conditions, between one analysis and the next.

All analytes were detected in ESI positive mode operating in scheduled ionization mode with a capillary voltage of 5500 V, nebulizer gas (air) at 40 psi, turbo gas (nitrogen) at 40 psi, and source temperature at 500 °C; the ionization widow was set from 1.5 min to 4.5 min during the chromatographic run. For all selected analytes, instrumental parameters, such as the declustering potential (DP), focusing potential (FP), entrance potential (EP), collision energy (CE), and cell exit potential (CXP), were tuned by infusion of each single standard methanolic solution (10 ng mL^−1^) at a flow rate of 10 μL min^−1^. For each analyte, two precursor ion/product ion transitions were chosen. The acquisition and quantification of the ion currents were performed in Multiple Reaction Monitoring (MRM) mode. The selected MRM transition for each target analyte, together with the main HPLC-MS/MS parameters, is shown in Appendix A. Data collection and processing were performed with Analyst 1.7.2 software, with quantification using Multiquant 3.0.3 software (Sciex).

### 3.5. Method Validation

The developed method was validated according to international guidelines provided by the FDA [35]. Several parameters were evaluated such as the LOD, LOQ, linearity, carry-over, matrix effect, recovery, accuracy, and precision.

#### 3.5.1. Calibrations Standards, Quality Controls, Linearity, and Carry-Over

Calibration standards (CSs) were prepared in MeOH at ten concentrations (0.1, 0.25, 0.5, 1, 2.5, 5, 10, 25, 50, 100 ng mL^−1^) to obtain a calibration curve. Quality controls (QCs) (n = 5, for each concentration) were prepared in lupin homogenate by spiking the analytical standards at target concentrations (25, 50, and 500 µg Kg^−1^). The QC_0_ samples were prepared in triplicate as blanks to determinate the endogenous LA concentration; for all QCs, the reported analytical procedure was then carried out.

Linearity was evaluated from iLOQs to 100 ng mL^−1^ for each analyte with the calibration curve described above, which was prepared on each validation day. The analyte response for the quantifier transition was normalized to the recovery and matrix effect, and calibration was achieved by linear least square regression. Carryover was assessed by injecting blank samples after high QC and high CS.

#### 3.5.2. Limit of Detection and Limit of Quantification

Being target analytes and possible endogenous compounds, LODs and LOQs were calculated by normalizing the instrumental limits (iLODs and iLOQs) with dilution factors, recoveries, and matrix effects. The iLODs and iLOQs were estimated at a signal-to-noise (S/N) ratio of 3 and 10, respectively.

#### 3.5.3. Accuracy and Precision

QCs were prepared for each concentration and analyzed in duplicate on three different days; accuracy was calculated as Bias% [42], taking into account the endogenous concentration. For this purpose, the following formula was used:$$\mathrm{Bias}\%=\frac{{\bar{x}}_{i}-\left(\mu +{\bar{x}}_{QC_{0}}\right)}{\left(\mu +{\bar{x}}_{QC_{0}}\right)}\times 100$$
where ${\bar{x}}_{i}$ is the mean measured concentration, *μ* corresponds to the theoretical concentration, and ${\bar{x}}_{QC_{0}}$ corresponds to the endogenous concentration for that sample set. Moreover, precision was calculated as the coefficient of variation (CV), by the following formula:$$\mathrm{CV}=\frac{\sigma }{{\bar{x}}_{i}}\times 100$$
where σ is the standard deviation and ${\bar{x}}_{i}$ is the mean measured concentration. For intraday precision, the values obtained for each individual day and for each QC class were taken into account. The values of all five days were considered for each QC group for the determination of intraday and interday precision.

#### 3.5.4. Recovery and Matrix Effect

Recovery (Rec%) and the matrix effect (ME%) were evaluated for each QC concentration by spiking lupin homogenate aliquots before dilution and after SPE extraction and processed with the same procedure; different reference mixtures were made for each QC.

The method used to calculate the recovery is based on the ratio of the area of the analyte in the sample spiked before the dilution (${\bar{A}}_{i}$) to the area of the sample (${\bar{A}}_{f}$). ME% was calculated as the ratio of the area of sample spiked after the SPE procedure (${\bar{A}}_{f}$), corrected by the area of the QC_0_ (${\bar{A}}_{e}$), to the reference mixture area (${\bar{A}}_{r}$):$$\mathrm{Rec}\%=\frac{{\bar{A}}_{i}}{{\bar{A}}_{f}}\times 100$$
$$\mathrm{ME}\%=\frac{{\bar{A}}_{f}-{\bar{A}}_{e}}{{\bar{A}}_{r}}\times 100$$

### 3.6. Statistical Analysis of the LA Content

The statistical analysis was performed with GraphPad Prism 9.1.0 (Dotmatics, Boston, MA, USA); the dataset was constituted by the concentration of each LA expressed as mg Kg^−1^, for a total of 12 observation (3 biological replicates for 4 lupin samples) and 6 variables (LAs); no missing data were present in the dataset. One-way ANOVA was performed to calculate the *p*-value, comparing both lupin samples of different sizes and also lupins with the same caliber but different farming origin, by multiple comparisons for each observed LA.

## 4. Conclusions

In this work, a rapid and reliable HPLC-MS/MS analytical method for the quantification of thirteen alkaloids in lupins was developed and validated according to FDA guidelines. The developed method is the first target LC-MS/MS with a large number of LAs and exhibits improved performance in term of sensitivity with respect to other target LC-MS/MS approaches. In addition, a fast and efficient extraction procedure was studied and substituted the classical approach consisting of acidification and alkalinization steps.

The presented method can represent a useful tool for studying differences in alkaloid contents and profiles among several lupin species, such as *L. albus* L., *L. angustifolius* L., *L. Luteus* L., and *L. mutabilis* L., as well as other plant genera such as *Baptisia*, *Thermopsis*, *Genista*, *Cytisus*, *Laburnum*, and *Sophora*.

Moreover, the analysis conducted on four different batches of raw *L. albus* L. seeds, originating from the Abruzzo region, showed a correlation with both growth and farming conditions, which can be helpful for a farmer to understand what type of seeds should be selected for human consumption and, consequently, for the most appropriate debittering process.

## Figures and Tables

**Figure 1 molecules-28-01531-f001:**
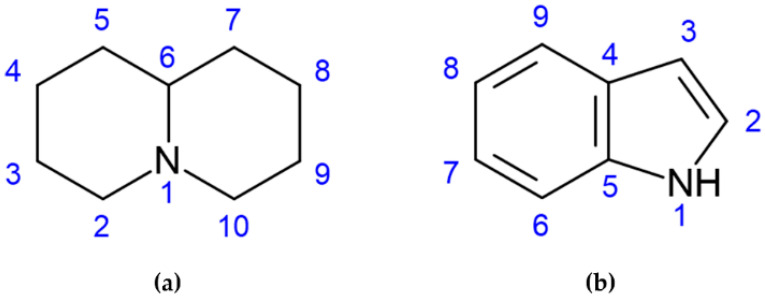
The skeleton structures of (**a**) quinolizidine and (**b**) indole alkaloids are shown.

**Figure 2 molecules-28-01531-f002:**
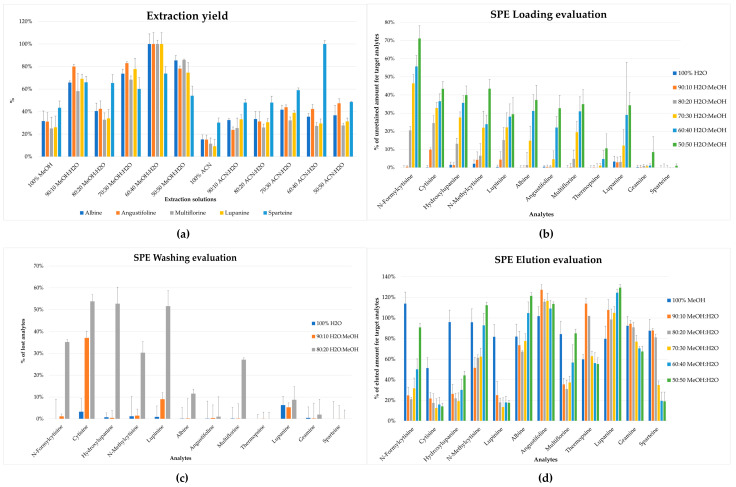
Evaluation of extraction and clean-up. (**a**) Extraction yield with different ratios of MeOH:H_2_O and ACN:H_2_O (*v*:*v*) for the most representative alkaloids in *L. albus* L.; all data were normalized (%) to the highest obtained value. (**b**) Loading tests; amount of retained compounds is reported. (**c**) Washing tests; amount of lost compounds is reported. (**d**) Elution tests; amount of eluted compounds is reported.

**Figure 3 molecules-28-01531-f003:**
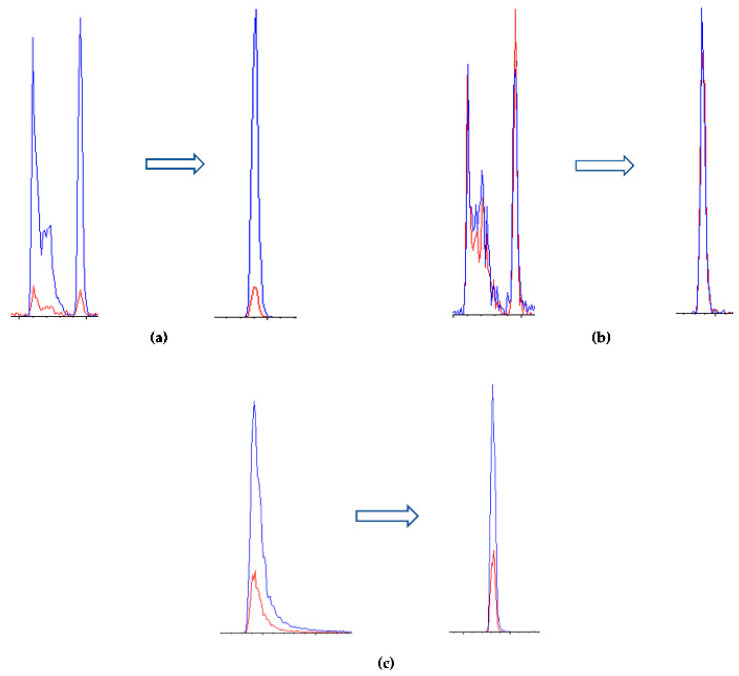
Improvement of the extracted ion current (XIC) of (**a**) N-methylcytisine, (**b**) cytisine, and (**c**) sparteine by comparison of the relative XICs resulting from the use of formic acid (left) and HFBA (right) in mobile phases using the C18-PFP column. All analytes were tested at a concentration of 20 ng mL^−1^.

**Figure 4 molecules-28-01531-f004:**
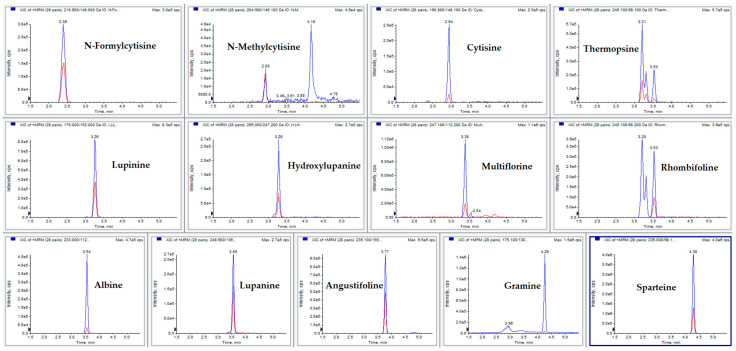
Chromatographic profiles of the selected analytes expressed as extracted ion currents (XICs), obtained by the developed MRM method; in the chromatogram, the chromatographic run of a standard solution at a concentration of 20 ng mL^−1^ is shown.

**Figure 5 molecules-28-01531-f005:**
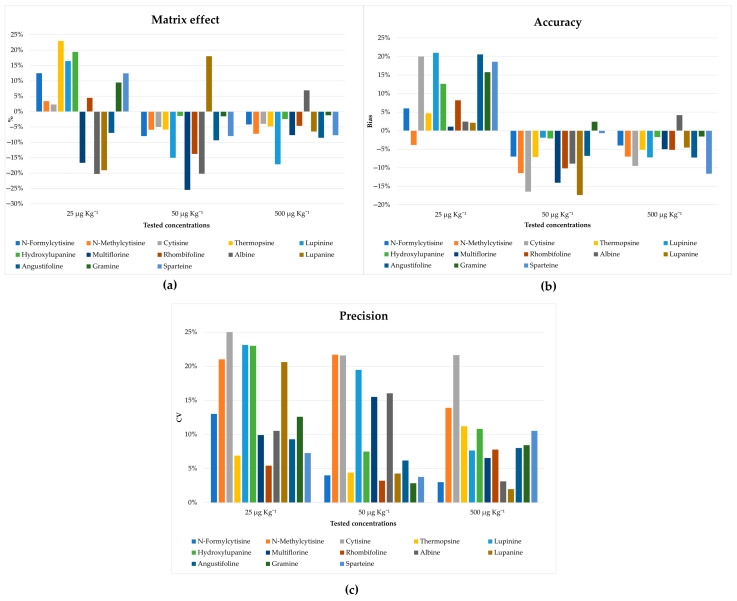
Validation parameters: (**a**) matrix effect reported as the suppression or enhancement of the signal, expressed as a percentage for all tested concentrations; (**b**) accuracy was reported as bias% for all tested concentrations; (**c**) precision was reported as CV for all tested concentrations.

**Figure 6 molecules-28-01531-f006:**
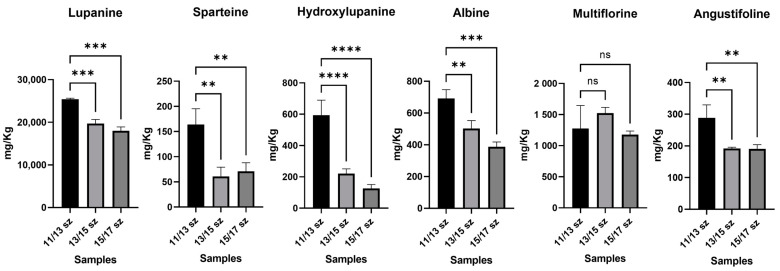
In these histogram plots, all 6 LAs detected in the raw lupin samples of different calibers were reported. The samples were grouped for each LA. The concentration of LAs was reported as mg Kg^−1^ of sample. Y axis title = Concentration (mg Kg^−1^). X axis title = Different caliber samples. (**) = *p*-value ≤ 0.01, (***) = *p*-value ≤ 0.005, (****) = *p*-value ≤ 0.001.

**Figure 7 molecules-28-01531-f007:**
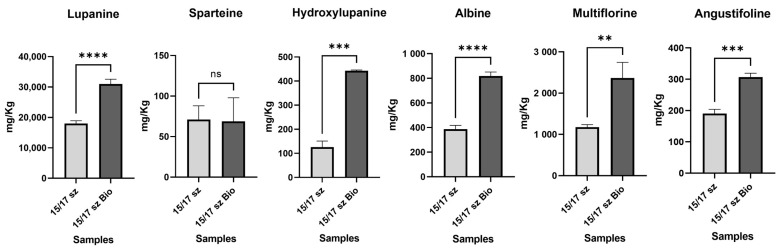
In this histogram plots, all 6 LAs detected in the conventional farming raw lupin samples and organic farming raw lupin samples were reported. The samples were grouped for each LA. The concentration of the LAs is reported as mg Kg^−1^ of sample. Y axis title = Concentration (mg Kg^−1^). X axis Title = Different farming conditions. (**) = *p*-value ≤ 0.01, (***) = *p*-value ≤ 0.005, (****) = *p*-value ≤ 0.001.

**Table 1 molecules-28-01531-t001:** Validation results: Linearity (Equation and R^2^), Limit of Detection (LOD), Limit of Quantification (LOQ).

Analytes	Equation	R²	LOD (µg Kg^−1^)	LOQ (µg Kg^−1^)
N-Formylcytisine	y = 1.838 × 10^4^x + (−4.403 × 10^3^)	0.998	1.5	5
N-Methylcytisine	y = 6.884x + 8.676 × 10^1^	0.996	7.5	25
Cytisine	y = 9.066 × 10^3^x + (−3.215 × 10^3^)	0.996	4.5	15
Thermopsine	y = 2.856 × 10^4^x + (−2.053 × 10^4^)	0.997	4	12.5
Lupinine	y = 2.966 × 10^4^x + (−1.054 × 10^4^)	0.990	2.5	9
Hydroxylupanine	y = 9.179 × 10^3^x + (−5.971 × 10^2^)	0.997	7	22.5
Multiflorine	y = 4.305 × 10^4^x + 6.417 × 10^3^	0.998	2	6.5
Rhombifoline	y = 1.192 × 10^4^x + 3.202 × 10^3^	0.991	4	12.5
Albine	y = 1.734 × 10^4^x + (−5.562 × 10^3^)	0.996	2	7
Lupanine	y = 1.067 × 10^4^x + 2.863 × 10^4^	0.997	2.5	7.5
Angustifoline	y = 3.260 × 10^4^x + 3.430 × 10^4^	0.995	2.5	9
Gramine	y = 4.805 × 10^4^x + 3.600 × 10^4^	0.990	4.5	15
Sparteine	y = 1.659 × 10^5^x + 1.157 × 10^4^	0.992	0.5	1

## Data Availability

Data are available from the corresponding author upon reasonable request.

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
