# Peer review of "Development of an HPLC-MS/MS Method for the Determination of Alkaloids in Lupins"

_molecules, 2023, doi:10.3390/molecules28041531_

Round 1
Reviewer 1 Report
Comments on the manuscript “Development of HPLC-MS/MS method for the determination 2 of alkaloids profile in lupins”. Various concerns to be addressed/discussed are as follows.
Introduction.
The full structures of the analytes should be given. Since few readers will be familiar with this set of compounds, I suggest including them in the main manuscript.
Section 2.2.
Regarding the optimization of the SPE, I have some doubts. In figure 2 the Y axis is not defined. For example, in figure 2b, I understand that the percentage not retained is shown, so the lower the value the better retention. This should be explained, and all axes have to be defined.
On the other hand, 100% MeOH is selected for elution. However, for some compounds, other more polar mixtures seem to be better. How is a compromise solution between all values ​​reached? Perhaps it would be worth ordering the compounds based on their polarities and justifying which conditions are better or worse based on this factor.
This kind of discussion could also be extended to the chromatographic separation, trying to justify the adequacy of the stationary phase based on the analyte features.
Section 2.3.
Authors could briefly discuss the ions of Q1 and Q3 depending on the analyte structures, at least for the most important compounds. Some structures with tentative fragmentations would be very illustrative.
Additional examples with other peaks can be added to complement Figure 3.
Section 2.5.
The title “real samples” is quite vague and it should be improved.
From the point of view of the target compounds, concentrations in Table S1 are much higher than the maximum values mentioned in line 73. Does this mean that the samples are very toxic? Can they be consumed? What applications would they have? The authors should discuss these aspects in the "Results and discussion" section.
Conclusions are a mere summary of the work. I expected some speculation on the advantages of the proposed method over others reported in the literature and some conclusions/recommendations on the farming practices. As far as I understand, the best products for human consumption should be those with lower alkaloid concentrations so that organic farming would be worse??
Table S1. Since samples have been analyzed in triplicate, the standard deviations should be given.
Minor comments:
Some paragraphs are quite short, even a single sentence. Please, try to make longer paragraphs by joining texts that are related.
Title: alkaloids is acting as an adjective so should be in singular e.g., “alkaloid profile”
Line 24: “determination” instead of “analysis”
Line 25: “able to detect” or “capable of detecting”
Line 28: “applied to”; check throughout the manuscript.
Line 66: “. Hence”
Line 101: “only a few”
Line 147: Please use the notation “MeOH:H2O 60:40 (v:v)” to express the composition of the hydro-organic solutions throughout the manuscript.
Line 192: “The two most intense…”
Line 201: “reaction” is not suitable.
Revise the LAs acronym. Sometimes is given as “Las”
Chemical compounds should be written in lower case. See, for instance, Albine, 13-Hydroxylupanine, Isolupanine, Angustifoline, etc.
Once each acronym is entered and defined, use it as is (e.g., in line 151, “Solid Phase Extraction (SPE)” has been already defined).
Line 157&158: “silica-based” and “polymeric-based”.
Figure 3 (caption): indicate the column.
Figure 4 (caption): the concentration of each analyte should be given. Thermopsine display tree peaks in both selected transitions. Do you have any explanation?
Section 2.4. More details about the method validation should be added (e.g., number of standards for calibration, number of replicates in the matrix effect, accuracy, precision, etc.), mentioned in the text or in the captions. I know is given in section 3.5 but some essential info could be duplicated here.
Do the FDA guidelines recommend that accuracy be expressed as error or as recovery (100%)? In my opinion, the recovery is a more common parameter.
Line 265: “; on the other hand, a batch of the same supplier underwent organic farming”.
Line 302: “as shown in”
Line 306: “was dependent on”
Line 336: Use “s” instead of “sec”.
Line 464: “determination” instead of “analysis”
Author Response
Comments on the manuscript “Development of HPLC-MS/MS method for the determination of alkaloids profile in lupins”. Various concerns to be addressed/discussed are as follows.
Introduction.
The full structures of the analytes should be given. Since few readers will be familiar with this set of compounds, I suggest including them in the main manuscript.
We agree with the reviewer suggestion, we reported the target compounds in supplementary materials in Tables S2a and Table S2b.
Section 2.2.
Regarding the optimization of the SPE, I have some doubts. In figure 2 the Y axis is not defined. For example, in figure 2b, I understand that the percentage not retained is shown, so the lower the value the better retention. This should be explained, and all axes have to be defined.
As suggested, both Figure 2 and text were improved, for the sake of clarity, the following sentences were inserted in paragraph (section 2.2)
“The first step was characterized by the selection of the loading phase, in order to obtain a satisfactory retention of the target analytes. Different ratios of H2O:MeOH (v:v) were tested (90:10, 80:20, 70:30, 60:40, 50:50 v:v) and the analytes were evaluated by comparing the area of the single unretained analyte to the area of a reference mixture with known concentration. The best conditions were obtained using H2O:MeOH 90:10 (v:v) that allowed the retention of the majority of the analytes in the cartridge, as shown in Figure 2b; the selected condition was then applied to the evaluation of washing and elution steps. Moreover, for the washing step, 100% H2O, 90:10 and 80:20 ratios (v:v) of H2O:MeOH were tested; H2O:MeOH 90:10 (v:v) was selected as washing solution, due to the reduced amount of lost compounds and, at the same time, low organic solvent was used. The latter was necessary to remove interferents affecting the ionization step in LC-MS/MS; this is reported in Figure 2c.
After the set-up of previously mentioned step, the elution conditions were tuned, testing MeOH and different ratios of MeOH:H2O (90:10, 80:20, 70:30, 60:40, 50:50 v:v). As reported in Figure 2d, 100% MeOH allowed a reliable recovery for all the target compounds from the cartridge (>50%). Despite the presence of H2O in different tested solvents improved the recovery of some analytes (i.e. lupanine, thermopsine and angustifoline), it negatively affected the elution of others (i.e. N-formylcytisine, N-methylcytisine, cytisine and lupinine); indeed MeOH was a good compromise to obtain a good recovery for all target analytes.”
In Figure 2 the y-axis was defined, and caption was changed as follows:
“Figure 2. Evaluation of extraction and clean-up. (a) Extraction yield with different ratios of MeOH:H2O and ACN:H2O (v:v) for the most representative alkaloids in L. albus L.; all the data were normalized (%) to the highest obtained value. (b) Loading tests; amount of retained compounds is reported. (c) Washing tests; amount of lost compounds is reported. (d) Elution tests; amount of eluted compounds is reported.”
On the other hand, 100% MeOH is selected for elution. However, for some compounds, other more polar mixtures seem to be better. How is a compromise solution between all values ​​reached? Perhaps it would be worth ordering the compounds based on their polarities and justifying which conditions are better or worse based on this factor.
The STRATA XL SPE cartridges used are supposed to retain the compounds not only according to their polarity; indeed a mixed mechanism, explained in datasheet of product, was reported in section 2.2, so the retention and the release of analytes depends by different factor, however the selection of the best conditions was a compromise regarding release of the analytes in elution step; according to experimental data, showed in figure 2d, MeOH present good and balanced performance for all analytes.
This kind of discussion could also be extended to the chromatographic separation, trying to justify the adequacy of the stationary phase based on the analyte features.
For sake of clarity, the HPLC paragraph was improved as follows:
“During the development of the LC method, the initial test was performed on a C18 stationary phase that gave a good separation of the analytes, except for more polar compounds which were poorly retained. On the other hand, the PFP stationary phase allowed a better retention for all the compounds, but provided a lower separation efficiency. The combined C18-PFP stationary phase appeared the most suitable for the analytes of interest, providing good performances for retention, peak shape and selectivity for all compounds; this combined stationary phase maintains the hydrophobic interaction and low bleed characteristics of C18 phases, with an improved retention of polar compounds due to PFP phase.
Several combinations of mobile phases containing different amounts of formic acid (ionization enhancer) were initially tested; good performances were obtained using water (H2O) with 0.1% formic acid as phase A and acetonitrile (ACN) as phase B, as reported in a previous work [39]. However, peak tailing was still present, particularly for sparteine, in addition peak splitting of cytisine and N-methylcytisine.
With the purpose to untangle these issues, formic acid was replaced with heptafluorobutyric acid (HFBA), that was added both in A and B phase at 0.1%; this improved both peak shape and ionization yield. Phase B was then replaced with ACN:MeOH 50:50 (v:v) 0.1% HFBA, in order to have a further enhancement of ionization. The resulting chromatographic profile was satisfactory as reported in Figure 3. The extracted ionic currents (XIC) for each analyte are reported in Figure 4.”
Section 2.3.
Authors could briefly discuss the ions of Q1 and Q3 depending on the analyte structures, at least for the most important compounds. Some structures with tentative fragmentations would be very illustrative.
Thanks for the useful comment, we have improved the discussion as suggested; the structures of the ion fragments were not showed in a figure as they are already reported in different papers by Hama et al, Carlier et al, Hwang et al, Petruczynik et al. the following sentence was reported in the revised version of the manuscript together with the reference listed.
“Tandem mass spectrometry parameters were carefully tuned by direct infusion of each analytical standard at concentration of 10 ng mL-1 and flow rate of 10 µL min-1. The most intense fragments were selected for each compound according to the literature [31-32,35,38] and tested during chromatographic run; then the two most intense MRM transition based on S/N ratio were selected as quantifier and qualifier.”
- Hama, J.R.; Strobel, B.W. Natural Alkaloids from Narrow-Leaf and Yellow Lupins Transfer to Soil and Soil Solution in Agricultural Fields. Environ. Sci. Eur. 2020, 32, 126, doi:10.1186/s12302-020-00405-7.
- Carlier, J.; Guitton, J.; Romeuf, L.; Bévalot, F.; Boyer, B.; Fanton, L.; Gaillard, Y. Screening Approach by Ultra-High Performance Liquid Chromatography-Tandem Mass Spectrometry for the Blood Quantification of Thirty-Four Toxic Principles of Plant Origin. Application to Forensic Toxicology. J. Chromatogr. B Anal. Technol. Biomed. Life Sci. 2015, 975, 65–76, doi:10.1016/j.jchromb.2014.10.028.
- Hwang, I.M.; Lee, H.W.; Lee, H.M.; Yang, J.S.; Seo, H.Y.; Chung, Y.J.; Kim, S.H. Rapid and Simultaneous Quantification of Five Quinolizidine Alkaloids in Lupinus Angustifolius l. And Its Processed Foods by UPLC-MS/MS. ACS Omega 2020, 5, 20825–20830, doi:10.1021/acsomega.0c01929.
- Petruczynik, A.; Wróblewski, K.; Misiurek, J.; Plech, T.; Szalast, K.; Wojtanowski, K.; Mroczek, T.; Szymczak, G.; Waksmundzka-Hajnos, M.; Tutka, P. Determination of Cytisine and N-Methylcytisine from Selected Plant Extracts by High-Performance Liquid Chromatography and Comparison of Their Cytotoxic Activity. Toxins 2020, 12, 557, doi:10.3390/toxins12090557
Additional examples with other peaks can be added to complement Figure 3.
As suggested by the reviewer, additional examples about peak shape improvements to amend chromatographic flaws (covering different aspects such as splitting, tailing, etc.) were added in figure 3.
Section 2.5.
The title “real samples” is quite vague and it should be improved.
As suggested by reviewer, the title of section 2.5 was modified as follows:
“2.5. Analysis of LAs in Lupinus albus L. samples”
From the point of view of the target compounds, concentrations in Table S1 are much higher than the maximum values mentioned in line 73. Does this mean that the samples are very toxic? Can they be consumed? What applications would they have? The authors should discuss these aspects in the "Results and discussion" section.
As reported in the introduction, Lupinus species, particularly the bitter ones, have an amount of LAs usually higher than other plants of the same family; for this reason, a treatment (generally washing for debittering) before their commercialization is always required [11]. The aim of this section was to test the proposed method, and the potential edibility of raw materials was not taken under consideration. However, it should be noted that the information of LAs profile could be useful for the farmer to understand which farming or growth condition could be taking into account to minimize the content of alkaloids in Lupins albus L.
- Narasinga Rao, M.S.; Rahma, E.H. Effect of Debittering Treatment on the Composition and Protein Components of Lupin See d (Lupinus Termis) Flour. J. Agric. Food Chem. 1984, 32, 1026–1030, doi:10.1021/jf00125a020.
This consideration was also added in section 2.5. as follows:
“According to these preliminary data, the different size and farming conditions seem to have a significant influence in the LAs content in L. albus L. This information in LAs content in Lupinus albus L. can be useful in farming leading to the selection of the conditions for industrial debittering process, a necessary step for lupins edibility.”
Conclusions are a mere summary of the work. I expected some speculation on the advantages of the proposed method over others reported in the literature and some conclusions/recommendations on the farming practices. As far as I understand, the best products for human consumption should be those with lower alkaloid concentrations so that organic farming would be worse??
The conclusion section was revised according to the reviewer suggestions:
“LAs represent a class of toxic secondary metabolites in plants, in particular in Lupinus spp., in which they act as a defence mechanism due to their strong bitter taste. They are considered very dangerous for human and animal health; for this reason, International Regulations require compliance with a maximum level of 200 mg Kg-1 of alkaloids in lupin flours and in the seeds for human consumption, that is possible after a proper debittering pre-treatment.
In this work, a rapid and reliable HPLC-MS/MS analytical method for the quantification of thirteen alkaloids in lupin was developed. The method was validated according to FDA guidelines. Considering the lack of works dealing with LAs determination in Lupinus spp, the method reported here is the first targeted LC-MS/MS with a large number of LAs and exhibits improved performances in term of sensitivity comparing to other target LC-MS/MS methods [35]. In addition, the determination of LAs in lupin samples was achieved via the development of a fast and efficient extraction procedure, compared to those present in literature consisting in two different steps of acidification and alkalinization [37].
The presented method can represent a useful tool for studying, with good sensitivity and reliability, the differences in alkaloid content and profile among several lupin species, such as L. albus L., L. angustifolius L., L. Luteus L., L. mutabilis L., but also in other plant genera such as Baptisia, Thermopsis, Genista, Cytisus, Laburnum, and Sophora.
Moreover, the analysis was conducted on four different batches of raw L. albus L. seeds originating in Abruzzo region, with different size and farming treatment. To the best of our knowledge, for the first time an analytical method based on targeted LC-MS/MS was developed and tested on this Lupinus spp., which is one of the most common in Europe, providing useful information about the alkaloids profile. The profile and correlation with both growth and farming conditions, can be helpful for the farmer to understand what type of seeds are more suitable for human consumption and, consequently, for the most appropriate debittering process [11].”
Table S1. Since samples have been analyzed in triplicate, the standard deviations should be given.
As suggested by reviewer the table S1 was modified with SD.
Minor comments:
Some paragraphs are quite short, even a single sentence. Please, try to make longer paragraphs by joining texts that are related.
The manuscript was revised following the suggestion.
Title: alkaloids is acting as an adjective so should be in singular e.g., “alkaloid profile”
Done
Line 24: “determination” instead of “analysis”
Done
Line 25: “able to detect” or “capable of detecting”
Done
Line 28: “applied to”; check throughout the manuscript.
Done
Line 66: “. Hence”
Done
Line 101: “only a few”
Done
Line 147: Please use the notation “MeOH:H2O 60:40 (v:v)” to express the composition of the hydro-organic solutions throughout the manuscript.
Done
Line 192: “The two most intense…”
Done
Line 201: “reaction” is not suitable.
Done
Revise the LAs acronym. Sometimes is given as “Las”
Done
Chemical compounds should be written in lower case. See, for instance, Albine, 13-Hydroxylupanine, Isolupanine, Angustifoline, etc.
Done
Once each acronym is entered and defined, use it as is (e.g., in line 151, “Solid Phase Extraction (SPE)” has been already defined).
Done
Line 157&158: “silica-based” and “polymeric-based”.
Done
Figure 3 (caption): indicate the column.
The caption of figure 3 was modified as follows:
“Figure 3. Improvement of extracted ion current (XIC) of a) N-methylcytisine, b) cytisine and c) sparteine, by comparison of the relative XICs resulting from the use of formic acid (left) and HFBA (right) in mobile phases using C18-PFP column. All the analytes were tested at a concentration of 20 ng mL-1.”
Figure 4 (caption): the concentration of each analyte should be given. Thermopsine display tree peaks in both selected transitions. Do you have any explanation?
The chromatographic profile of the LAs was an important aspect of this paper, we finely tuned the LC parameters for almost target analytes. Anyway, Thermopsine display tree peaks in both selected transitions; the LC development solved the peak splitting phenomena as explained in section 2.3, but for thermopsine the issue was not completely resolved, anyway performance of the method in term of precision and accuracy were not influenced by this issue. Moreover, one of the three mentioned peaks (rt = 3.53 min) was referred to rhombifoline and not to thermopsine, anyway good peak separation was performed, and we were able to distinguish the two mentioned above compounds in term of both chromatographic and mass spectrometry separation.
Moreover, the concentration of analytes is given in the caption.
Section 2.4. More details about the method validation should be added (e.g., number of standards for calibration, number of replicates in the matrix effect, accuracy, precision, etc.), mentioned in the text or in the captions. I know is given in section 3.5 but some essential info could be duplicated here.
As suggested by reviewer some information were added in the section 2.4 as follows:
“Validation parameters were then evaluated, according to FDA guidelines [36]. All QCs samples were prepared in lupins homogenate as reported in section 3.5.1; briefly, different QCs were prepared at different concentrations (25, 50, 500 µg Kg-1) and rep-licates (n=5) were tested. Moreover, 10-points calibration curve was prepared in MeOH (0.1, 0.25, 0.5, 1, 2.5, 5, 10, 25, 50, 100 ng mL-1) for linearity test.”
Do the FDA guidelines recommend that accuracy be expressed as error or as recovery (100%)? In my opinion, the recovery is a more common parameter.
Different parameters were tested in order to validate the proposed method such as recovery, matrix effect accuracy in according with the FDA guidelines in which recovery or error were recognise as parameters for accuracy; in other hand Thompson et al. suggested to give information about recovery and error if corrected values were used; for this reason, we preferred to express accuracy as bias of values which were corrected with recovery. The complete recovery results, used to correct the value, were also added in Supplementary Materials, in particular in Figure S1.
- Thompson, M.; Ellison, S.L.R.; Fajgelj, A.; Willetts, P.; Wood, R. Harmonised Guidelines for the Use of Recovery Information in Analytical Measurement (Technical Report). Pure Appl. Chem. 1999, 71, 337-348, doi:10.1351/pac199971020337
Line 265: “; on the other hand, a batch of the same supplier underwent organic farming”.
Done
Line 302: “as shown in”
Done
Line 306: “was dependent on”
Done
Line 336: Use “s” instead of “sec”.
Done
Line 464: “determination” instead of “analysis”
Done

Reviewer 2 Report
The manuscript “Development of HPLC-MS/MS method for the determination of alkaloids profile in lupinsClean up procedure” is an exciting work. After having read it, only a few doubts have to be answered.
A complete clean-up procedure has been optimized, but no bibliographic references have been mentioned. In lines 157 and 163, two different types of the cartridge are mentioned. I guess authors must have based their selection of the kind of cartridge on some information before publishing. You must cite.
Additionally, the same in the optimization of the HPLC-MS/MS. Why do authors start for mobile phase water with formic acid 0.1 % as phase A and CAN as phase B?. There must be some previous feedback that has to be cited.
You mention that “The presented analytical method was validated following the Food and Drug Admin- 128 istration (FDA) guidelines”. It will be a good idea to briefly mention the acceptable precision, accuracy and matrix effect % acceptable for the range of concentrations found in the samples. For me, a precision higher of 10 % is not a good result even though the degree of concentration is ppbs.
Author Response
Reviewer 2
The manuscript “Development of HPLC-MS/MS method for the determination of alkaloids profile in lupins Clean up procedure” is an exciting work. After having read it, only a few doubts have to be answered.
A complete clean-up procedure has been optimized, but no bibliographic references have been mentioned. In lines 157 and 163, two different types of the cartridge are mentioned. I guess authors must have based their selection of the kind of cartridge on some information before publishing. You must cite.
We selected the cartridge due to different type of interaction, in order to evaluate the most suitable condition for the clean-up procedure; to improve the clarity of the manuscript the following sentences were added:
“The clean-up step was tuned by developing an efficient SPE procedure, to provide suitable recoveries and the minimization of the matrix effect for the analytes. For this purpose, two different SPE cartridges were initially evaluated, considering different types of stationary phases, such as a silica-based Strata C18-E (55 μm particle size) and a polymeric Strata XL (100 μm particle size); the different stationary phases were selected for their different mechanism of interaction with the target analytes, to achieve a satisfactory clean-up procedure.”
Additionally, the same in the optimization of the HPLC-MS/MS. Why do authors start for mobile phase water with formic acid 0.1 % as phase A and CAN as phase B?. There must be some previous feedback that has to be cited.
The initial HPLC condition were in line with our previous work on carbazole alkaloids, anyway further tuning of chromatographic condition was necessary; for better understanding the manuscript was modified as follows:
“Several combinations of mobile phases containing different amounts of formic acid (ionization enhancer) were initially tested; good performances were obtained using water (H2O) with 0.1% formic acid as phase A and acetonitrile (ACN) as phase B, as reported in a previous work [39]. However, peak tailing was still present, particularly for sparteine, in addition peak splitting of cytisine and N-methylcytisine.”
- Viteritti, E.; Oliva, E.; Eugelio, F.; Fanti, F.; Palmieri, S.; Bafile, E.; Compagnone, D.; Sergi, M. Analysis of Carbazole Alkaloids in Murraya Koenigii by Means of High Performance Liquid Chromatography Coupled to Tandem Mass Spectrometry with a Predictive Multi Experiment Approach. J. Chromatogr. Open 2022, 2, 100055, doi:10.1016/j.jcoa.2022.100055.
You mention that “The presented analytical method was validated following the Food and Drug Administration (FDA) guidelines”. It will be a good idea to briefly mention the acceptable precision, accuracy and matrix effect % acceptable for the range of concentrations found in the samples. For me, a precision higher of 10 % is not a good result even though the degree of concentration is ppbs.
Thanks to the referee for the suggestion, different acceptability range was given by the FDA in correlation of concentration, in particular, for the tested concentration, the accepted CV for the FDA was ±30%, for sake of clarity the reference was added.
- Horwitz, W. Evaluation of Analytical Methods Used for Regulation of Foods and Drugs. Anal. Chem. 1982, 54, 67–76, doi:10.1021/ac00238a002.

Reviewer 3 Report
The authors develop and validate an analytical HPLC-MS/MS methodology for the quantification of alkaloids present in lupin matrices. The results demonstrate that the method is sensitive and robust.
Abstract: A good context is presented, the objective of the manuscript is clear, some relevant results are reported, and a general conclusion is presented.
Keywords: The journal allows up to 10 keywords, a higher number could improve the visibility of your manuscript if it is published.
Introduction: The subject matter is adequately contextualized, the knowledge gap is well defined and the importance of the scientific contribution of the manuscript is clearly defined. the main objective of the work is described and the main conclusions are highlighted.
Results and discussions: The results are clear and convincing, and the analytical quality of the data supports the authors' discussion.
1) Line 137, the word solubilization is not suitable, since it implies the use of surfactants. The extraction is based on the solubility of the alkaloids.
Materials and Methods: The methods are well described, demonstrating high technical/scientific quality.
Conclusions: It is recommended that the conclusions be modified, since much of the text is redundant, due to the fact that sections already described in both results and methods are presented.
In most of the text, results are presented; conclusions require propositions based on the results.
Author Response
Reviewer 3
The authors develop and validate an analytical HPLC-MS/MS methodology for the quantification of alkaloids present in lupin matrices. The results demonstrate that the method is sensitive and robust.
Abstract: A good context is presented, the objective of the manuscript is clear, some relevant results are reported, and a general conclusion is presented.
Keywords: The journal allows up to 10 keywords, a higher number could improve the visibility of your manuscript if it is published.
As suggested by reviewer the number of keywords were improved up to 5.
Introduction: The subject matter is adequately contextualized, the knowledge gap is well defined and the importance of the scientific contribution of the manuscript is clearly defined. the main objective of the work is described and the main conclusions are highlighted.
Results and discussions: The results are clear and convincing, and the analytical quality of the data supports the authors' discussion.
1) Line 137, the word solubilization is not suitable, since it implies the use of surfactants. The extraction is based on the solubility of the alkaloids.
As suggested, the sentences were modified as follows:
“In the literature, most of the studies focused on a specific alkaloid extraction procedure, based on their dissolution in acidic solution and subsequent alkalinization, followed by a further purification by means of liquid-liquid extraction or SPE [37]”
Materials and Methods: The methods are well described, demonstrating high technical/scientific quality.
Conclusions: It is recommended that the conclusions be modified, since much of the text is redundant, due to the fact that sections already described in both results and methods are presented.
In most of the text, results are presented; conclusions require propositions based on the results.
As suggested by the reviewer, the conclusion was revised, and some aspects were defined as follows:
“LAs represent a class of toxic secondary metabolites in plants, in particular in Lupinus spp., in which they act as a defence mechanism due to their strong bitter taste. They are considered very dangerous for human and animal health; for this reason, International Regulations require compliance with a maximum level of 200 mg Kg-1 of alkaloids in lupin flours and in the seeds for human consumption, that is possible after a proper debittering pre-treatment.
In this work, a rapid and reliable HPLC-MS/MS analytical method for the quantification of thirteen alkaloids in lupin was developed. The method was validated according to FDA guidelines. Considering the lack of works dealing with LAs determination in Lupinus spp, the method reported here is the first targeted LC-MS/MS with a large number of LAs and exhibits improved performances in term of sensitivity comparing to other target LC-MS/MS methods [35]. In addition, the determination of LAs in lupin samples was achieved via the development of a fast and efficient extraction procedure, compared to those present in literature consisting in two different steps of acidification and alkalinization [37].
The presented method can represent a useful tool for studying, with good sensitivity and reliability, the differences in alkaloid content and profile among several lupin species, such as L. albus L., L. angustifolius L., L. Luteus L., L. mutabilis L., but also in other plant genera such as Baptisia, Thermopsis, Genista, Cytisus, Laburnum, and Sophora.
Moreover, the analysis was conducted on four different batches of raw L. albus L. seeds originating in Abruzzo region, with different size and farming treatment. To the best of our knowledge, for the first time an analytical method based on targeted LC-MS/MS was developed and tested on this Lupinus spp., which is one of the most common in Europe, providing useful information about the alkaloids profile. The profile and correlation with both growth and farming conditions, can be helpful for the farmer to understand what type of seeds are more suitable for human consumption and, consequently, for the most appropriate debittering process [11].”

Reviewer 4 Report
- After reading the manuscript I have the following remarks and suggestions:
1. According to the IUPAC recommendation, analytical results should be presented in the form: x ± expanded uncertainty
2. Results in Table S1 should be supported with expanded uncertainty or RSD.
3. Error bars representing the value of expanded uncertainty or RSD value should be added to all experimental points in Figure...
4. In the univariate analysis, the factors must be fixed very close to the optimized values. In this case, the authors did not specify the fixed factors in Figures (2b, 2c, 2d). Please specify the fixed value when optimizing each one.
5. That's important to understand the huge gap in the results of Figures 2c and 2d !! So, please specify the fixed value when optimizing each one.
6. For the selection of columns in the HPLC analysis, have you already evaluated the three columns? if yes, please add the results, and if no and you just conclude from the feature, no need to add the third paragraph in section 2.3. Just report that you used the following column for the following reasons.
7. It is also surprising that MeCN only causes peak tailing while MeCN: MeOH, 50: 50 improves the peak shape !!! MeCN has higher elution power than MeOH!!
8. The HPLC/MS-MS section has many unimportant details, please paraphrase and keep the important notices, especially if the experiments were not performed!!
9. Y-axis labels should be added in figures 2 and 5
10. Page 7, line 250, it is not about robustness!!! robustness is another validation parameter and should be evaluated in a different way!!
11. All specific information related to the used standards should be included (chemical name, identifiers (i.e., CAS), supplier, purity, chemical structure, prepared concentrations, etc.).
12. The structures and logp for the studied alkaloids should be added to understand the separation mechanisms
13. I am surprised that a solution of 60:40 147 MeOH: H2O yield 100% for nearly all the analytes except 1!!!!!
14. The conclusion should indicate the main conclusions or interpretations (summarized), not the introduction as written by the authors. I see it very very long!!!!!!!!!!!! please revise and paraphrase
15. A comparison between the results of the developed method and the literature should be added.
Author Response
Reviewer 4
After reading the manuscript I have the following remarks and suggestions:
- According to the IUPAC recommendation, analytical results should be presented in the form: x ± expanded uncertainty
- Results in Table S1 should be supported with expanded uncertainty or RSD.
As suggested by the reviewer in comments 1 and 2 the Table S2 was modified.
- Error bars representing the value of expanded uncertainty or RSD value should be added to all experimental points in Figure...
The error bars were added to Figure 2 as suggested by reviewer, moreover error bars in figure 7 were homogenized.
- In the univariate analysis, the factors must be fixed very close to the optimized values. In this case, the authors did not specify the fixed factors in Figures (2b, 2c, 2d). Please specify the fixed value when optimizing each one.
The development of SPE procedure followed specific workflow for testing of more suitable condition for the clean-up; after the evaluation of each step (loading, washing and elution) the previous evaluated proportion of solvent was maintained to test the other next step; For sake of clarity the paragraph was improved as follows:
“The first step was characterized by the selection of the loading phase, in order to obtain a satisfactory retention of the target analytes. Different ratios of H2O:MeOH (v:v) were tested (90:10, 80:20, 70:30, 60:40, 50:50 v:v) and the analytes were evaluated by comparing the area of the single unretained analyte to the area of a reference mixture with known concentration. The best conditions were obtained using H2O:MeOH 90:10 (v:v) that allowed the retention of the majority of the analytes in the cartridge, as shown in Figure 2b; the selected condition was then applied to the evaluation of washing and elution steps. Moreover, for the washing step, 100% H2O, 90:10 and 80:20 ratios (v:v) of H2O:MeOH were tested; H2O:MeOH 90:10 (v:v) was selected as washing solution, due to the reduced amount of lost compounds and, at the same time, low organic solvent was used. The latter was necessary to remove interferents affecting the ionization step in LC-MS/MS; this is reported in Figure 2c.
After the set-up of previously mentioned step, the elution conditions were tuned, testing MeOH and different ratios of MeOH:H2O (90:10, 80:20, 70:30, 60:40, 50:50 v:v). As reported in Figure 2d, 100% MeOH allowed a reliable recovery for all the target compounds from the cartridge (>50%). Despite the presence of H2O in different tested solvents improved the recovery of some analytes (i.e. lupanine, thermopsine and angustifoline), it negatively affected the elution of others (i.e. N-formylcytisine, N-methylcytisine, cytisine and lupinine); indeed MeOH was a good compromise to obtain a good recovery for all target analytes.”
- That's important to understand the huge gap in the results of Figures 2c and 2d !! So, please specify the fixed value when optimizing each one.
The SPE step development was tuned in according to step by step evaluation, so the experiment about elution step was performed after the washing step test, in which the proportion of H2O:MeOH 90:10 was chosen in according to the % of lost amount and using at least a bit portion of MeOH, in order to minimize the matrix effect on target analytes. The comments of the reviewer had highlighted lack of clarity in this section which was improved as describe above.
- For the selection of columns in the HPLC analysis, have you already evaluated the three columns? if yes, please add the results, and if no and you just conclude from the feature, no need to add the third paragraph in section 2.3. Just report that you used the following column for the following reasons.
According to reviewer suggestion, more details about using C18 and PFP were added to the manuscript as follows:
“During the development of the LC method, the initial test was performed on a C18 stationary phase that gave a good separation of the analytes, except for more polar compounds which were poorly retained. On the other hand, the PFP stationary phase allowed a better retention for all the compounds, but provided a lower separation efficiency. The combined C18-PFP stationary phase appeared the most suitable for the analytes of interest, providing good performances for retention, peak shape and selectivity for all compounds; this combined stationary phase maintains the hydrophobic interaction and low bleed characteristics of C18 phases, with an improved retention of polar compounds due to PFP phase.”
- It is also surprising that MeCN only causes peak tailing while MeCN: MeOH, 50: 50 improves the peak shape !!! MeCN has higher elution power than MeOH!!
The peak shape for all analytes was improved using HFBA in mobile phase. Due to this improvement, we were able to add MeOH in mobile phase without having tailing issues during the chromatographic run. MeOH was added to improve the ionization of the compounds. Due to lack of clarity the manuscript was modified as follows:
“With the purpose to untangle these issues, formic acid was replaced with heptafluorobutyric acid (HFBA), that was added both in A and B phase at 0.1%; this improved both peak shape and ionization yield. Phase B was then replaced with ACN:MeOH 50:50 (v:v) 0.1% HFBA, in order to have a further enhancement of ionization. The resulting chromatographic profile was satisfactory as reported in Figure 3. The extracted ionic currents (XIC) for each analyte are reported in Figure 4.”
- The HPLC/MS-MS section has many unimportant details, please paraphrase and keep the important notices, especially if the experiments were not performed!!
For sake of clarity the paragraph was improved as follows:
“During the development of the LC method, the initial test was performed on a C18 stationary phase that gave a good separation of the analytes, except for more polar compounds which were poorly retained. On the other hand, the PFP stationary phase allowed a better retention for all the compounds, but provided a lower separation efficiency. The combined C18-PFP stationary phase appeared the most suitable for the analytes of interest, providing good performances for retention, peak shape and selectivity for all compounds; this combined stationary phase maintains the hydrophobic interaction and low bleed characteristics of C18 phases, with an improved retention of polar compounds due to PFP phase.”
- Y-axis labels should be added in figures 2 and 5
The caption of figure 2 and 5 were improved according to the reviewer suggestion as follows, moreover the caption were modified:
“Figure 2. Evaluation of extraction and clean-up. (a) Extraction yield with different ratios of MeOH:H2O and ACN:H2O (v:v) for the most representative alkaloids in L. albus L.; all the data were normalized (%) to the highest obtained value. (b) Loading tests; amount of retained compounds is reported. (c) Washing tests; amount of lost compounds is reported. (d) Elution tests; amount of eluted compounds is reported.”
“Figure 5. Validation parameters: a) matrix effect reported as suppression or enhancement of the signal, expressed as percentage for all tested concentrations; b) accuracy was reported as bias% for all tested concentrations; c) precision was reported as CV for all tested concentrations.”
- Page 7, line 250, it is not about robustness!!! robustness is another validation parameter and should be evaluated in a different way!!
We agree with the reviewer, the sentence was modified as follows:
“Accuracy and precision of the method were also tested, which, as shown in Figure 5b and 5c, are in line with the guidance provided by the FDA [40]. Moreover, no carry-over effects were observed during the tests.”
- All specific information related to the used standards should be included (chemical name, identifiers (i.e., CAS), supplier, purity, chemical structure, prepared concentrations, etc.).
According to the reviewer suggestion different information about used standard were implemented in Table S2a and Table S2b, instead prepared concentrations of standard solution were reported in manuscript as follow:
“The working standard mixtures were prepared by appropriate dilution in MeOH (10, 1, 0.01 µg mL-1) and stored at −20°C.”
- The structures and logp for the studied alkaloids should be added to understand the separation mechanisms
As suggested by the reviewer the structure and logP of the target analytes were reported in TableS2a and TableS2b.
- I am surprised that a solution of 60:40 147 MeOH: H2O yield 100% for nearly all the analytes except 1!!!!!
In the paper, different amounts of inorganic and organic solvents were tested to evaluate the most suitable condition for maximizing the extraction of the target compounds, so the experimental data are presented in Figure 2a to indicate the motivation in choice of extraction mixture; probably the gap in compounds extraction was caused by the interactions between the analytes and the matrix components that may affect the extraction efficiency.
- The conclusion should indicate the main conclusions or interpretations (summarized), not the introduction as written by the authors. I see it very very long!!!!!!!!!!!! please revise and paraphrase
- A comparison between the results of the developed method and the literature should be added.
As suggested by reviewer in comments 14 and 15, the conclusion was revised and summarized, moreover the comparison with the literature was improved, as follows:
“LAs represent a class of toxic secondary metabolites in plants, in particular in Lupinus spp., in which they act as a defence mechanism due to their strong bitter taste. They are considered very dangerous for human and animal health; for this reason, International Regulations require compliance with a maximum level of 200 mg Kg-1 of alkaloids in lupin flours and in the seeds for human consumption, that is possible after a proper debittering pre-treatment.
In this work, a rapid and reliable HPLC-MS/MS analytical method for the quantification of thirteen alkaloids in lupin was developed. The method was validated according to FDA guidelines. Considering the lack of works dealing with LAs determination in Lupinus spp, the method reported here is the first targeted LC-MS/MS with a large number of LAs and exhibits improved performances in term of sensitivity comparing to other target LC-MS/MS methods [35]. In addition, the determination of LAs in lupin samples was achieved via the development of a fast and efficient extraction procedure, compared to those present in literature consisting in two different steps of acidification and alkalinization [37].
The presented method can represent a useful tool for studying, with good sensitivity and reliability, the differences in alkaloid content and profile among several lupin species, such as L. albus L., L. angustifolius L., L. Luteus L., L. mutabilis L., but also in other plant genera such as Baptisia, Thermopsis, Genista, Cytisus, Laburnum, and Sophora.
Moreover, the analysis was conducted on four different batches of raw L. albus L. seeds originating in Abruzzo region, with different size and farming treatment. To the best of our knowledge, for the first time an analytical method based on targeted LC-MS/MS was developed and tested on this Lupinus spp., which is one of the most common in Europe, providing useful information about the alkaloids profile. The profile and correlation with both growth and farming conditions, can be helpful for the farmer to understand what type of seeds are more suitable for human consumption and, consequently, for the most appropriate debittering process [11].”

Round 2
Reviewer 1 Report
The paper is ready to be published.
Author Response
Reviewer 1
The paper is ready to be published.
Thanks to reviewer for the comment.

Reviewer 4 Report
The authors have replied to all the comments, however, I still believe that the conclusion needs to be improved. It is still very long. It must be a short quantitative summary to the results and must not contain any references.
Author Response
Reviewer 4
The authors have replied to all the comments, however, I still believe that the conclusion needs to be improved. It is still very long. It must be a short quantitative summary to the results and must not contain any references.
In according with the reviewer suggestion, the conclusion section was improved. and summarized as follows:
“In this work, a rapid and reliable HPLC-MS/MS analytical method for the quantification of thirteen alkaloids in lupin was developed and validated according to FDA guidelines. The developed method is the first target LC-MS/MS with a large number of LAs and exhibits improved performances in term of sensitivity with respect to other target LC-MS/MS approaches. In addition, a fast and efficient extraction procedure was studied, and substituted the classical consisting in acidification and alkalinization steps.
The presented method can represent a useful tool for studying differences in alkaloid content and profile among several lupin species, such as L. albus L., L. angustifolius L., L. Luteus L., L. mutabilis L., but also in other plant genera such as Baptisia, Thermopsis, Genista, Cytisus, Laburnum, and Sophora.
Moreover, the analysis conducted on four different batches of raw L. albus L. seeds, originating in Abruzzo region, showed a correlation with both growth and farming conditions, which can be helpful for the farmer to understand what type of seeds should be selected for human consumption and, consequently, for the most appropriate debittering process.”